# SOLVING CONTINUAL LEARNING VIA PROBLEM DECOMPOSITION

## ABSTRACT

This paper is concerned with *class incremental learning* (CIL) in continual learning (CL). CIL is the popular continual learning paradigm in which a system receives a sequence of tasks with different classes in each task and is expected to learn to predict the class of each test instance without given any task related information for the instance. Although many techniques have been proposed to solve CIL, it remains to be highly challenging due to the difficulty of dealing with catastrophic forgetting (CF). This paper starts from the first principle and proposes a novel method to solve the problem. The definition of CIL reveals that the problem can be decomposed into two probabilities: within-task prediction probability and task-id prediction probability. This paper proposes an effective technique to estimate these two probabilities based on the estimation of feature distributions in the latent space using incremental PCA and Mahalanobis distance. The proposed method does not require a memory buffer to save replay data and it outperforms strong baselines including replay-based methods.[1]

## 1    INTRODUCTION

Continual learning (CL) is a learning problem where a system learns and accumulates knowledge over time without forgetting the previous knowledge (Chen & Liu, 2018). The key challenge is the catastrophic forgetting (CF), which is a phenomenon that the system corrupts the learned knowledge in the past in learning a new task (McCloskey & Cohen, 1989). This paper focuses on the challenging CL setting of *class incremental learning* (CIL) (Rebuffi et al., 2017) in the offline (or batch) mode. In this setting, the system learns a sequence of classification tasks incrementally, where each task arrives with all its training data of a set of classes. The resulting classifier can identify the class of a test instance among all the classes learned in the process with no task information provided. The other popular setting of CL is *task incremental learning* (TIL), which builds a separate model for each task and in testing, the test instance together with the task-id that the test instance belongs to are provided so that the system can use the model of the specific task to classify the instance.

Existing approaches to CIL can be grouped into several categories. Regularization (Kirkpatrick et al., 2017) or distillation (Li & Hoiem, 2016) tries not to change the parameters or knowledge that are important to old tasks when learning the new task. Replay/memory-based approaches (Rebuffi et al., 2017) save some old data and use them jointly with the new task data to learn the new task and to preserve/adjust the old knowledge. Parameter isolation approaches (Serra et al., 2018) expand the network or mask out the important parameters for old tasks (see Sec. 2 for more details). Our approach is entirely different and is derived directly from the definition of the CIL setting.

**Definition:** *Class incremental learning* (CIL) learns a sequence of tasks $1, ..., t$, where each task $i$ has a training data $\boldsymbol{D}^i = \{(\boldsymbol{x}^i_j, y^i_j)\}_{j=1}^{n^i}$ with $\boldsymbol{x}^i_j \in \boldsymbol{X}^i$ (input space) and $y^i_j \in \boldsymbol{Y}^i$ (class label space). The class labels of tasks are disjoint, $\boldsymbol{Y}^i \cap \boldsymbol{Y}^k = \emptyset$ for any $i \neq k$ [2]. Let $\boldsymbol{X} = \cup_{i=1}^t \boldsymbol{X}^i$ and $\boldsymbol{Y} = \cup_{i=1}^t \boldsymbol{Y}^i$. The goal is to learn a function $f : \boldsymbol{X} \to \boldsymbol{Y}$ to predict the class label of test case $\boldsymbol{x}$.

---

[1]The code is included in the Supplementary Material.

[2]In (Bang et al., 2021), tasks are considered to have shared classes. For instance, the system receives two datasets $\boldsymbol{D}^1$ and $\boldsymbol{D}^2$ consisting of classes $\{y_1, y_2\}$ and $\{y_1, y_3, y_4\}$, respectively. We define task 1 and 2 consisting of $\{y_1, y_2\}$ and $\{y_3, y_4\}$, respectively, and consider the samples of shared label $y_1$ as additional training data for task 1. This work does not consider this learning scenario. We leave it for our future work.

As tasks have disjoint classes, the CIL probability of a sample $\boldsymbol{x}$ having the $j$th class label $y_j^i$ of task $i$ can be decomposed into two probabilities,

$$\mathbf{P}(y_j^i|\boldsymbol{x}) = \mathbf{P}(y_j^i|\boldsymbol{x}, i)\mathbf{P}(i|\boldsymbol{x}), \tag{1}$$

The decomposition implies that there are two probabilities that define the CIL probability. The first probability on the right-hand-side (RHS) is the *within-task prediction* (WTP) probability (or *intra-task prediction* probability) and the second probability on the RHS is the *task-id prediction* (TIP) probability (or *inter-task prediction* probability). Thus, a system makes a correct CIL prediction if it produces accurate within-task and task-id predictions.

We note that the WTP probability is exactly the prediction probability in a TIL problem. However, in TIL, the task-id is given in inference or testing. Thus, to solve the CIL problem, one can learn like TIL and then design a mechanism to predict the task-id to which the test instance belongs. Some existing works have taken this approach (Rajasegaran et al., 2020; Abati et al., 2020), but they perform poorly because their task-id predictors are very weak. However, these papers did not propose Eq. 1. We will discuss these and other related works in the related work section. In fact, the WTP probability can be improved from that given by a TIL system too.

This paper proposes a novel technique to estimate the two probabilities and an exemplar-free CIL system, called **EWT** (*Estimation of **W**TP and **T**IP probabilities*). EWT makes use of the highly effective hard-attention masking method HAT (Serra et al., 2018) for TIL to learn feature extractor for each task. HAT has almost no forgetting for TIL as it masks out the parameters and neurons learned for previous tasks. This ensures that the estimated probabilities are robust and are not affected by forgetting in incremental learning. Although we could directly use the probability of each class produced from each task as WTP probability, this approach is sub-optimal. We propose a generative approach to improve the estimation by considering possible noisy and/or out-of-distribution samples. This is done by fine-tuning the task classifiers using generated pseudo feature representations for each class. The generation is done based on the Gaussian distributions in the latent feature space. The Gaussian distribution for each class is estimated incrementally using *incremental Principle Component Analysis* (iPCA). The TIP probability is estimated using Mahalanobis distance.

Our experiments demonstrate the effectiveness of the proposed method EWT using a pre-trained transformer network that does not have information leak, i.e., it is trained using the ImageNet data with all classes that are similar to the classes in the experiment datasets removed. Both our system and the baselines fix the transformer and train adapter modules inserted at each transformer layer (Houlsby et al., 2019). The experimental results shows that EWT outperforms the recent state-of-the-art baselines by large margins, including replay-based approaches.

## 2 RELATED WORK

Numerous techniques have been proposed for CL. We consider the five most relevant categories: exemplar-free, replay-based, generative methods, network-expansion, and parameter isolation. Exemplar-free methods (saving no previous task data) often use regularization (Kirkpatrick et al., 2017; Zhu et al., 2021), knowledge distillation (Li & Hoiem, 2016), or orthogonal projection (Zeng et al., 2019) to preserve previous important parameters (Zenke et al., 2017; Wang et al., 2022). Our method is also exemplar-free and our CF prevention is based on task masking (Serra et al., 2018).

Replay-based CL has been widely studied in CIL. Different saving mechanisms (Rebuffi et al., 2017; Liu et al., 2020b; Bang et al., 2022), replay strategies (Aljundi et al., 2019), and regularizations (Lopez-Paz & Ranzato, 2017; Castro et al., 2018; Chaudhry et al., 2018; Buzzega et al., 2020; Chaudhry et al., 2021) have been used. The goal of these methods is to balance the plasticity and stability using the saved samples of previous tasks (Liu et al., 2021; Yan et al., 2021). Our method does not save any samples and it also performs much better than recent replay-based methods.

Generative methods (Shin et al., 2017; Ostapenko et al., 2019; Ayub & Wagner, 2021) build generators to generate pseudo-replay data to prevent forgetting. Lesort et al. (2018) studied the difficulties of the generative approach. We do not generate pseudo-replay samples similar to the raw data and thus do not have its problems. Our method generates feature vectors rather than raw data. Liu et al. (2020a) and Zhu et al. (2021) also generate feature vectors. They use the generated features for distilling knowledge of previous tasks. However, we estimate the distributions of features to fine-tune the classifier and to compute task-id probability rather than for knowledge distillation.

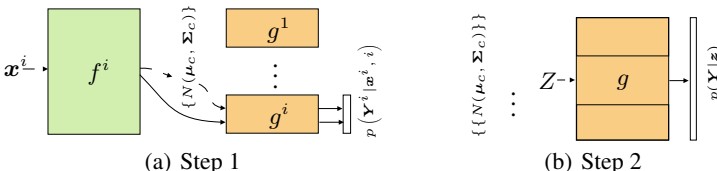

(a) Step 1          (b) Step 2

Figure 1: Overview of the training process for task $i$. The dashed lines indicate that the gradient flow is blocked while the solid lines indicate gradients are computed along the lines. (a) The network is trained in a multi-head manner (one per task), in which task-specific parameters are trained based on hard-attention and used to effectively eliminate the interference between tasks. During training, the system dynamically estimates the distributions $N(\boldsymbol{\mu}_c(f^i(\boldsymbol{x}^i)), \boldsymbol{\Sigma}_c(f^i(\boldsymbol{x}^i)))$ of feature vectors $f^i(\boldsymbol{x}^i)$. The estimated distributions are used to generate pseudo feature vectors, and the network is jointly trained with the pseudo feature vectors and the training data of task $i$. Since the network $g^i \circ f^i$ is trained to minimize the loss on both training data and the pseudo feature vectors, this process encourages feature vectors of $f^i$ to follow the desired distribution. (b) Given the distributions, the system fine-tunes a classifier $g$ created by joining the multi-head classifiers $g^k$, $k \leq i$. The fine-tuning is done using pseudo feature vectors generated from the distributions.

Network expansion methods (Rusu et al., 2016; Mehta et al., 2021; Yan et al., 2021) expand the network to preserve old parameters. IBP-WF (Mehta et al., 2021) uses global weight factors for knowledge sharing and a Bayesian non-parametric approach for network expansion. It first finds the task-id in testing using Gaussian distributions and then uses the task-id to select the correct model for prediction. Our method is different as we do not expand the network or find the task-id, but directly compute the CIL probability. DER (Yan et al., 2021) expands the network and also does pruning to reduce the network size. Our method does not expand the network.

Another popular branch in continual learning is parameter isolation, which trains a set of task specific parameters. The methods are mostly designed for task incremental learning (TIL) as they require the task-id of each test instance to choose the correct task specific parameters. We leverage the hard attention masking (HAT) (Serra et al., 2018) to prevent forgetting. However, our method is for CIL unlike the original HAT. Although there are attempts (von Oswald et al., 2020; Rajasegaran et al., 2020; Abati et al., 2020; Henning et al., 2021) to use the parameter isolation methods for CIL problem, they do not tackle CIL by problem decomposition problematically as we do. These methods are much weaker than ours (see Sec. 4).

Using a pre-trained model (e.g., BERT, GPT-3, ViT, DeiT, or CLIP) has been a standard practice for CL in natural language processing (Ke et al., 2021). For image data, Ostapenko et al. (2022) studied using pre-trained models as foundation models for CL. SLDA (Hayes & Kanan, 2020) fixes the pre-trained feature extractor and fine-tunes the classifier. L2P (Wang et al., 2022) trains a prompt pool with a fixed feature extractor and Wu et al. (2022) fine-tunes replicate layers of a pre-trained model. Our method EWT also leverages a pre-trained feature extractor, but we use adapters in the fixed feature extractor and trains only the adapters to learn new knowledge. Using the same pre-trained model, our method outperforms SLDA and L2P by a large margin (see Sec. 4).

Different CIL settings have been studied as well. Blurry task (or task-free) is studied in online CIL (Buzzega et al., 2020; Bang et al., 2022), where the tasks boundaries are not clear as tasks change gradually. Our method is an offline method, where tasks are disjoint. As noted in footnote 2, we split tasks by unseen classes rather than by datasets. We leave training with additional samples of previous tasks for our future work in the online CL setting.

## 3   PROPOSED METHOD

An overview of the training process of the proposed method is illustrated in Fig. 1. Learning a new task $i$ involves two steps. Step 1 focuses on training the feature extractor. Specifically, it trains the task network $g^i \circ f^i$ using both the training data $\boldsymbol{X}^i$ of task $i$ and the pseudo feature vectors $\boldsymbol{Z}$ generated from the Gaussian distribution $N(\boldsymbol{\mu}_c(f^i(\boldsymbol{X}^i)), \boldsymbol{\Sigma}_c(f^i(\boldsymbol{X}^i))$ of feature vectors for each class $c$ in the task (see Fig. 1(a)). The Gaussian distribution is dynamically and incrementally estimated

using *incremental Principle Component Analysis* (iPCA) during training (see Sec. 3.1.1). Since the network is jointly trained with the training data and the generated features, which also depend on the values of feature extractor $f^i$, this encourages the feature vector to follow the distribution. This step has little forgetting as the training is done based on the hard-attention mechanism in Serra et al. (2018), which can protect/mask the parameters learned from previous tasks (see Sec. 3.1.2). Note that although $f^i$'s are task specific but they are all learned in the same network and there are a lot of parameter sharing. Step 2 computes the two probabilities in Eq. 1 based on the trained feature extractors $f^i$'s and the Gaussian distribution for each class.

## 3.1 STEP 1: TASK TRAINING

We first discuss the detailed training process to learn a task $i$ in step 1, which performs two functions: (i) estimating the distribution of feature vectors for each class in a task using *incremental Principal Component Analysis* (iPCA), and (ii) training the feature extractor with hard attention masking in (Serra et al., 2018) to prevent interference or CF in learning task $i$ using the training data $\boldsymbol{X}^i$ and the generated feature vectors $\boldsymbol{Z}$ based on the incrementally estimated distributions on the fly. As we explain above, we use $\boldsymbol{Z}$ in the step 1 training because we want to produce better distributions, which will be used in step 2.

We train the network for task $i$ by minimizing the loss

$$\mathcal{L}_{ce} = -\frac{1}{2|\boldsymbol{B}|} \left( \sum_{(\boldsymbol{x},y) \in \boldsymbol{B}} \log p(y|\boldsymbol{x}, i) + \sum_{(\boldsymbol{z},y) \in \boldsymbol{Z}} \log p(y|\boldsymbol{z}, i) \right), \tag{2}$$

where the first $p$ on the right is the softmax output $g^i(f^i(\boldsymbol{x}))$, the second $p$ is the softmax output $g^i(\boldsymbol{z})$, $\boldsymbol{B}$ is a batch of training data, and $\boldsymbol{Z}$ is a batch of pseudo feature vectors generated from the Gaussian distributions $\{N(\boldsymbol{\mu}_c, \boldsymbol{\Sigma}_c)\}$ for each class $c$ in task $i$. The following sub-sections describe how to estimate the distributions of feature vectors and how to train the feature extractor without CF.

### 3.1.1 INCREMENTAL ESTIMATION OF GAUSSIAN DISTRIBUTIONS OF FEATURE VECTORS

We use multivariate Gaussian distribution to approximate the feature distribution for each class. The challenges in estimating the distribution of features in CL are: (i) the statistics $(\boldsymbol{\mu}_c, \boldsymbol{\Sigma}_c)$ need to be updated incrementally at each batch on the evolving feature extractor as using the whole data to recompute the statistics after training is computationally demanding and (ii) saving statistics of the Gaussian distributions is expensive as the feature vectors have a high dimension $d$. Therefore, we take ideas from the algorithms developed for incremental Principal Component Analysis (iPCA) to approximate the covariance. In iPCA, only a few ($k \ll d$) principal vectors are saved and updated dynamically at each batch without using any previous data. Since the following discussion is about estimating the distribution of feature vectors of each class, we remove the class indicator $c$ for simplicity in notation. Likewise, we also remove task index $i$.

Suppose we have seen $n$ training samples so far. Denote the $n$ samples of a class by $\boldsymbol{X}$ and the the feature vectors by $f(\boldsymbol{X}) = \boldsymbol{Z} = [\boldsymbol{z}_1, \cdots, \boldsymbol{z}_n]$ from the feature extractor $f$ while minimizing Eq. 2. Denote the sample mean by $\boldsymbol{\mu} = \sum \boldsymbol{z}_j / n$. Suppose that the system receives a new batch $\boldsymbol{X}_{new}$ of $m$ instances. We obtain the feature vectors $\boldsymbol{Z}_{new} = [\boldsymbol{z}_{n+1}, \cdots, \boldsymbol{z}_{n+m}]$, and update the mean by

$$\tilde{\boldsymbol{\mu}} = (n\boldsymbol{\mu} + m\boldsymbol{\mu}_{new})/(n+m), \tag{3}$$

where $\boldsymbol{\mu}_{new}$ is the sample mean of the new batch. Denote the singular value decomposition (SVD) of the centered feature by $\boldsymbol{U}\boldsymbol{\Lambda}\boldsymbol{V}^T \overset{\text{svd}}{=} [\boldsymbol{Z} - \boldsymbol{\mu}]$, where $T$ is the transpose symbol. We approximate the covariance with $k$ leading eigenvectors and eigenvalues as follows

$$(n-1)\boldsymbol{\Sigma} \approx \boldsymbol{U}_k \boldsymbol{\Lambda}_k^2 \boldsymbol{U}_k^T. \tag{4}$$

For simplicity, we denote the reduced matrices $\boldsymbol{U}_k$ by $\boldsymbol{U}$ and $\boldsymbol{\Lambda}_k$ by $\boldsymbol{\Lambda}$ as the following discussions are based on the reduced matrices. Based on (Ross et al., 2008), the SVD can be updated for a new set of data $\boldsymbol{Z}_{new}$ as $\tilde{\boldsymbol{U}}\tilde{\boldsymbol{\Lambda}}\tilde{\boldsymbol{V}}^T \overset{\text{svd}}{=} [\sqrt{n-1}\boldsymbol{U}\boldsymbol{\Lambda} \quad \boldsymbol{K}]$ given the block matrix $\boldsymbol{K} = [\boldsymbol{Z}_{new} - \boldsymbol{\mu}_{new} \quad \sqrt{nm/(n+1)}(\boldsymbol{\mu}_{new} - \boldsymbol{\mu})]$, and we obtain

$$(n+m-1)\boldsymbol{\Sigma} \approx \tilde{\boldsymbol{U}}\tilde{\boldsymbol{\Lambda}}^2\tilde{\boldsymbol{U}}^T. \tag{5}$$

The derivation for Eq. 5 is given in Appendix D. The statistics for the Gaussian distribution of class $c$ is estimated dynamically with $k$ ($<< d$) eigenpairs, and pseudo feature vectors can be drawn from the estimated distribution to train the classifier.

### 3.1.2 Hard Attention Masking

In training the network $g^i \circ f^i$ using the data of task $i$ and the generated pseudo feature vectors, we employ the hard attention mask (Serra et al., 2018) to prevent forgetting in the feature extractor.

The hard attention mask $\boldsymbol{a}_l^i$ is a trainable pseudo binary 0-1 vector at each layer $l$ of task $i$. It is element-wise multiplied to the output of the layer as $\boldsymbol{a}_l^i \otimes \boldsymbol{h}_l$ and blocks (for value of 0) or unblocks (for value of 1) the information flow from neurons of adjacent layers. Neurons with value 1 are important for the task and thus need to be protected while neurons with value 0 are not necessary for the task and can be freely modifed without affecting other tasks.

More specifically, we modify the gradients of parameters that are important in performing the previous tasks $(1, \cdots, i-1)$ during training task $i$ so the important parameters for previous tasks are unaffected. The gradient of parameter $w_{kj,l}$ at $k$th row and $j$th column of layer $l$ is modified as

$$\nabla w'_{kj,l} = \left( 1 - \min \left( a_{k,l}^{<i}, a_{j,l-1}^{<i} \right) \right) \nabla w_{kj,l}, \tag{6}$$

where $a_{k,l}^{<i}$ is an accumulated attentions over previous tasks and is 1 if the hard attention of neuron $k$ at layer $l$ is ever used by any previous task $< i$ (see (Serra et al., 2018) for details).

To encourage parameter sharing and sparsity in the number of activated masks, a regularization is introduced as $\mathcal{L}_r = \sum_{l,k} a_{k,l}^i (1 - a_{k,l}^{<i}) / \sum_{l,k} (1 - a_{k,l}^{<i})$. The final objective to train a comprehensive task network without forgetting is

$$\mathcal{L} = \mathcal{L}_{ce} + \mathcal{L}_r, \tag{7}$$

where $\mathcal{L}_{ce}$ is the cross-entropy loss in Eq. 2.

## 3.2 Step 2: Computing the Two Probabilities in Eq. 1

We now discuss how to compute the within-task prediction (WTP) probability and the task-id prediction (TIP) probability.

### 3.2.1 Computing the WTP Probability

We could use the softmax probability of each class in a task as the within task prediction (WTP) probability for the class. However, this method is not the best for computing the probability (see the experiment section) because those samples that may be outliers, noises, or other hard-to-classify cases are unlikely to get accurate probabilities, which also affect the probabilities of those samples that are easy to classify. We propose to consider possible out-of-distribution (OOD) samples in each task. However, we do not have OOD data for each task to use in learning the task. Since we have already computed the distributions of feature representations for each class in step 1, for each task we could use the generated data from the distributions of the other tasks as the OOD data for the task. Although we could consider the generated data from previous tasks as OOD data when training a new task, we cannot use the generated data from a later task to update the model of an earlier task because we no longer have the data of the earlier task and even if we can use the generated data of this earlier task, updating its feature extractor can cause serious forgetting because the feature extractors for different tasks share many parameters in the hard attentions (Serra et al., 2018).

We propose a simpler method. We build and fine-tune (see below) a separate linear classifier (with one input layer and one output layer) considering the classes from all tasks learned so far using the generated feature vectors for each class in each task from the feature distribution of the class estimated in step 1. The advantage of this approach is that using a single combined model/classifier we can consider the OOD data for all tasks because for the classes of a task, the classes of all other tasks can be considered as OOD data for the task.

After training each task $i$ in step 1, we have the set of distributions $\{\{N(\boldsymbol{\mu}_c, \boldsymbol{\Sigma}_c)\}_c\}_i$ of features of each class $c$ of the task $i \le t$. We then fine-tune a combined classifier $g$, which is the classifier cre-

ated by joining the parameters of each task's classifier $g^i$, using pseudo feature vectors $\boldsymbol{Z}$ generated from the distributions. This is illustrated in Fig. 1(b).

Note that in step 1, each task network $g^i \circ f^i$ is trained independently without considering the other task networks. In this step, we consider the outputs of all the tasks together and fine-tune the combined classifier $g$. We minimize the cross entropy loss

$$\mathcal{L}_{ce} = -\frac{1}{|\boldsymbol{Z}|} \sum_{(\boldsymbol{z},y) \in \boldsymbol{Z}} \log p(y|\boldsymbol{z}) \tag{8}$$

where the probability is computed using the softmax $[g^1(\boldsymbol{z}); \cdots ; g^t(\boldsymbol{z})]$. The WTP probability of class $c$ (which is our class label $y_j^i$ in Eq. 1) is

$$\mathbf{P}(c|\boldsymbol{x},i) = \mathrm{softmax}(g^i(f^i(\boldsymbol{x}))). \tag{9}$$

### 3.2.2 COMPUTING THE TIP PROBABILITY

We now compute the task-id prediction (TIP) probability for a given test sample $\boldsymbol{x}$. We make use of the distance between the feature vector $f^i(\boldsymbol{x})$ of $\boldsymbol{x}$ and a distribution $N(\boldsymbol{\mu}_c, \boldsymbol{\Sigma}_c)$ of features estimated by the training data. This has been used as an effective measure for OOD detection (Lee et al., 2018). We define the covariance of the distribution of task $i$ as $\boldsymbol{\Sigma}^i = \sum_{c \in C^i} \boldsymbol{\Sigma}_c / |C^i|$, where $C^i$ is the set of classes of task $i$ and $\boldsymbol{\Sigma}_c$ is the covariance matrix computed by the method discussed in Sec. 3.1.1 with all the principal components. We discard the class covariance after the computation to save memory. Given a set of distributions $\{N(\boldsymbol{\mu}_c, \boldsymbol{\Sigma}^i)\}_{c \in C^i}$ of task $i$ and a test instance $\boldsymbol{x}$, we define the following score of the feature $f^i(\boldsymbol{x})$,

$$s^i(\boldsymbol{x}) = 1/ \max_c \{MD(f^i(\boldsymbol{x}); \boldsymbol{\mu}_c, \boldsymbol{\Sigma}^i)\}, \tag{10}$$

where $MD$ is the Mahalanobis distance of sample $\boldsymbol{x}$ to the distribution $N(\boldsymbol{\mu}_c, \boldsymbol{\Sigma}^i)$. The higher the value, the further away the sample is from the distributions of task $i$.

Finally, the TIP probability for task $i$ is defined as,

$$\mathbf{P}(i|\boldsymbol{x}) = s^i(\boldsymbol{x})/ \sum_k s^k(\boldsymbol{x}), \tag{11}$$

Eq. 11 is justified as a sample that is closer to a distribution is more like to belong to the distribution.

## 4 EXPERIMENT

**Baselines.** We compare the proposed EWT with 11 baselines among which five are exemplar-free (i.e., saving no previous task data) methods and six are replay-based methods. The exemplar-free methods are: **HAT** (Serra et al., 2018), **OWM** (Zeng et al., 2019), **SLDA** (Hayes & Kanan, 2020), **PASS** (Zhu et al., 2021), and **L2P** (Wang et al., 2022). For the multi-head method HAT, we make prediction by taking $\arg \max$ over the concatenated logits from each task network as it works the best among all the considered prediction methods (refer to Appendix C for details). The replay methods are: **iCaRL** (Rebuffi et al., 2017), **A-GEM** (Chaudhry et al., 2018), **EEIL** (Castro et al., 2018), **DER++** (Buzzega et al., 2020), **HAL** (Chaudhry et al., 2021), and **DER** without pruning (Yan et al., 2021). We could not run (Wu et al., 2022) as no code was released. We also do not include the existing parameter isolation methods that deal with CIL problems as they are very weak [3].

**Datasets.** We use four popular continual learning benchmark datasets. **(1). CIFAR10** (Krizhevsky & Hinton, 2009). This is an image classification dataset consisting of 60,000 color images of size 32x32, among which 50,000 are training data and 10,000 are testing data. It has 10 different classes.

---

[3]HyperNet (von Oswald et al., 2020) and PR (Henning et al., 2021) find the task-id via an entropy function and SupSup (Wortsman et al., 2020) finds it via gradient update. They then make a within-task prediction. SupSup, PR, and iTAML (Rajasegaran et al., 2020) assume the test instances come in batches and all samples in a batch belong to one task. When tested per sample on ResNet-18, HyperNet, SupSup, PR and iTAML achieve 22.4, 11.8, 45.2 and 33.5 on 10 tasks of CIFAR100, respectively, which are much lower than 51.4 of the baseline iCaRL. CCG (Abati et al., 2020) and IBP-WF (Mehta et al., 2021) do not provide code.

**(2). CIFAR100** (Krizhevsky & Hinton, 2009). This dataset consists of 50,000 training images and 10,000 testing images with 100 classes. Each image is colored and of size 32x32. **(3). Tiny-ImageNet** (Le & Yang, 2015). This classification dataset has 200 classes with 500 training images of size 64x64 per class. The validation data has 50 samples per class. Since no label is provided for the test data, we use the validation set for testing as in (Zhu et al., 2021). **(4). ImageNet380**. We randomly selected 380 classes from the 389 classes, which are the remaining classes after removing those classes similar to those in CIFAR and Tiny-ImageNet from the original 1,000 classes of the full ImageNet data (Russakovsky et al., 2015) for pre-training (see below). This dataset has about 1,300 color images per class. Similar to Tiny-ImageNet above, we use the validation set (50 images per class) for testing as its original test data has no label.

**Backbone Architecture.** We use the backbone architecture of transformer DeiT-S/16 (Touvron et al., 2021). We initially pre-train the network using 611 classes of ImageNet after removing 389 classes which are similar or identical to the classes of CIFAR and Tiny-ImageNet. To leverage the strong performance of the pre-trained model while adapting to new knowledge, we fix the feature extractor and append trainable adapter modules of fully-connected networks with one hidden layer at each transformer layer (Houlsby et al., 2019) except SLDA and L2P [4]. The number of neurons in each hidden layer is 64 for CIFAR10 and 128 for other datasets. Note that ***all baselines and our method use the same architecture and the same pre-training model for fairness*** as using a pre-trained model improves the performance (Ostapenko et al., 2022) (e.g., DER improves from 65.2 to 73.3 on 10 tasks of CIFAR100 with pre-training on the same transformer architecture).

Note that we do not use the pre-trained models like CLIP (Radford et al., 2021) or others trained using the full ImageNet data due to information leak both in terms of features and class labels because our experiment data have been used in training these pre-trained models. This leakage can seriously affect the results. For example, the L2P system using the pre-training model trained using the full ImageNet data performs extremely well, but after those overlapping classes are removed in pre-training, its performances drop greatly. In Table 1, we can see that it is in fact quite weak.

**Training Details.** For saving eigenpairs, we follow the existing memory budget strategy in the replay-based method (Chaudhry et al., 2019) for fairness. We fix the total number of eigenpairs saved in the CL process. After learning a new task, the system discards $q$ eigenpairs with the smallest eigenvalues from each class of the previous tasks to accommodate $k$ eigenpairs of each newly learned class. This strategy maintains $k$ eigenvectors and the corresponding eigenvalues per class in the budget. Denote the budget size by $|\mathcal{M}|$.

For CIFAR10, we split the 10 classes into 5 tasks with 2 classes per task. The size of the hidden layer for the adapter module is 64 and the number of eigenpairs is 10 per class. We refer the experiment as C10-5T. The memory budget size $|\mathcal{M}|$ for eigenpairs is 100.

For CIFAR100, we conduct two experiments. We split the 100 classes into 10 and 20 tasks, where each task has 10 classes and 5 classes, respectively. We refer the experiments as C100-10T and C100-20T. We choose $|\mathcal{M}| = 1,000$ for both experiments.

For Tiny-ImageNet, we conduct two experiments. We split the 200 classes into 5 tasks with 40 classes per task and 10 tasks with 20 classes per task. We refer the experiments as T-5T and T-10T, respectively. We save 2,000 eigenpairs in total for both experiments.

For ImageNet380, we split the classes into 10 tasks with 38 classes per task and save 7,600 eigenpairs in total. We refer the experiment to I380-10T.

For all the experiments of our system, we find a good set of learning rates and the number of epochs via validation data made of 10% of the training data. We train our model for 15 epochs and use SGD with batch size of 128 and with momentum value 0.9 for step 1. For the experiments of CIFAR10 and CIFAR100, we use learning rate of 0.05 and 0.01, respectively. For Tiny-ImageNet and ImageNet, we use learning rate 0.005. We train the classifier in step 2 for 35 epochs with SGD with the same batch size and learning rate as step 1. Following the random class order protocol of the existing methods (Rebuffi et al., 2017; Yan et al., 2021), we randomly generate 5 different class orders for each experiment and report the average accuracy over the 5 random orders. For replay-based baselines, we follow Rebuffi et al. (2017). The systems use the memory buffer of size 200 for

---

[4]For SLDA and L2P, we follow the original papers. SLDA fine-tunes only the classifier with a fixed feature extractor and L2P trains learnable prompts.

Table 1: Average classification accuracy after the final task. '-XT' means X number of tasks. Our system EWT and all baselines used the pre-trained network. The last column shows the average of each method over all datasets and experiments. We highlight the best results in each column in bold.

| Method | C10-5T | C100-10T | C100-20T | T-5T | T-10T | I380-10T | Average |
|--------|--------|----------|----------|------|-------|----------|---------|
| HAT | 79.36±5.16 | 68.99±0.21 | 61.83±0.62 | 65.85±0.60 | 62.05±0.55 | 71.20±0.99 | 68.21 |
| OWM | 41.69±6.34 | 21.39±3.18 | 16.98±4.44 | 24.55±2.48 | 17.52±3.45 | 0.26±0.00 | 20.40 |
| SLDA | **88.64**±0.05 | 67.82±0.05 | 67.80±0.05 | 57.93±0.05 | 57.93±0.06 | 65.78±0.05 | 67.65 |
| PASS | 86.21±1.10 | 68.90±0.94 | 66.77±1.18 | 61.03±0.38 | 58.34±0.42 | 65.27±1.24 | 67.75 |
| L2P | 73.59±4.15 | 61.72±0.81 | 53.84±1.59 | 59.12±0.96 | 54.09±1.14 | 47.89±3.24 | 58.38 |
| iCaRL | 87.55±0.99 | 68.90±0.47 | 69.15±0.99 | 53.13±1.04 | 51.88±2.36 | 62.23±0.66 | 65.47 |
| A-GEM | 56.33±7.77 | 25.21±4.00 | 21.99±4.01 | 30.53±3.99 | 21.90±5.52 | 30.38±10.02 | 31.06 |
| EEIL | 82.34±3.13 | 68.08±0.51 | 63.79±0.66 | 53.34±0.54 | 50.38±0.97 | 63.37±0.49 | 63.55 |
| DER++ | 84.63±2.91 | 69.73±0.99 | 70.03±1.46 | 55.84±2.21 | 54.20±3.28 | 66.53±2.36 | 66.83 |
| HAL | 84.38±2.70 | 67.17±1.50 | 67.37±1.45 | 52.80±2.37 | 55.25±3.60 | 64.83±2.60 | 65.30 |
| DER | 86.79±1.20 | 73.30±0.58 | **72.00**±0.57 | 59.57±0.89 | 57.18±1.40 | 69.19±1.36 | 69.67 |
| EWT | 87.60±1.77 | **74.15**±0.40 | 71.06±0.35 | **66.16**±0.28 | **64.59**±0.17 | **74.30**±0.46 | **72.98** |

CIFAR10, 2,000 for CIFAR100 and Tiny-ImageNet, and 7,600 for ImageNet and save a set of raw training samples according to the saving strategy in the respective original papers [5]. For the other baselines, we follow the experiment setups as reported in their official papers.

**Evaluation Metrics.** We use two metrics: average classification accuracy (ACA) and average forgetting rate. ACA after the last task $t$ is $\mathcal{A}_t = \sum_{i=1}^{t} A_i^t/t$, where $A_i$ is the accuracy of the model on task $i$th data after learning task $t$. The average forgetting rate after task $t$ is $\mathcal{F}_t = \sum_{i=1}^{t-1} A_i^i - A_i^t$. This is also referred as backward transfer in other literature (Lopez-Paz & Ranzato, 2017). We report the incremental classification accuracy (ICA) and ACA at each task in Appendix E.

### 4.1 RESULTS AND COMPARISON

**Average Classification Accuracy.** Tab. 1 shows the average classification accuracy after the final task. The last column Average indicates the average performance of each method over the 6 experiments. Our proposed method EWT performs the best on average. We achieve 72.98% while the best baseline (DER) achieves 69.67%. The performance gap is even larger when we compare it with non-replay based methods. The best exemplar-free method is HAT and it achieves 68.21% on average, which is much lower than our method.

The baselines SLDA and L2P are proposed to leverage a strong pre-trained feature extractor in the original papers. SLDA freezes the feature extractor and only fine-tunes the classifier. It performs well for the simple experiment C10-5T but is significantly poorer than our EWT on other experiments. This is because the fixed feature extractor does not adapt to new knowledge. Our method updates the feature extractor via adapter modules to new knowledge and it is able to learn more complex problems. L2P trains a set of prompt embeddings. In the original paper, it uses a feature extractor that was pre-trained with ImageNet-21k which already includes the classes of the continual learning evaluation datasets. When we remove the classes similar to the datasets used in CL, its performance drops dramatically (58.38% on average over the 6 experiments) and much poorer than our method EWT (72.98% on average).

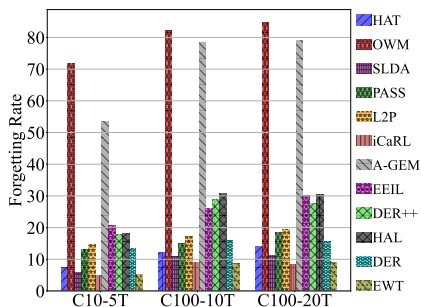

Figure 2: Average forgetting rate. The lower the rate, the better the method is.

**Average Forgetting Rate (Backward Transfer).** We compare the forgetting rate of each system after learning the last task in Fig. 2. The forgetting rates of the proposed method EWT are 5.26, 8.75,

---

[5]Note that we save eigenvectors where each vector is in dimension of 384 whereas the replay-based methods save raw inputs. For a memory of size $|\mathcal{M}|$ and dataset with $C$ classes, our method and the replay methods save $k = |\mathcal{M}|/C$ pairs and raw inputs, respectively, after the last task. Thus, for C100-10T, EWT takes 384K elements for the eigenpairs while replay methods consumes 6.1M elements. Refer to Appdendix B for details.

Table 2: The average classification accuracy by different variants of the proposed technique. The variant S1 indicates the model after step 1. The variant S1 + S2 indicates the model after step 2 and XX + TIP indicates the model with TIP of Sec. 3.2.2 applied at prediction.

|  | C10-5T | C100-10T | C100-20T | T-5T | T-10T | I380-10T |
|---|---|---|---|---|---|---|
| S1 | 74.00±3.93 | 67.76±0.49 | 59.91±0.66 | 64.35±0.61 | 60.48±0.24 | 70.66±0.91 |
| S1 + S2 | 84.05±2.24 | 71.22±0.44 | 66.11±0.72 | 65.25±0.39 | 61.95±0.47 | 71.95±0.79 |
| S1 + TIP | 82.83±2.78 | 71.84±0.65 | 66.84±0.50 | 65.97±0.74 | 63.49±0.37 | 73.45±1.14 |
| S1 + S2 + TIP (EWT) | 87.60±1.77 | 74.15±0.40 | 71.06±0.35 | 66.16±0.28 | 64.59±0.17 | 74.30±0.46 |

Table 3: The accuracy performance and the number of saved eigenpairs on C100-10T. $|\mathcal{M}| = m$ indicates that a total of $m$ eigenvectors are saved with their corresponding eigenvalues.

| $|\mathcal{M}| =$ | 100 | 500 | 1,000 | 1,500 | 2,000 |
|---|---|---|---|---|---|
| C100-10T | 73.41±0.06 | 74.10±0.62 | 74.15±0.40 | 74.25±0.06 | 73.87±0.22 |

and 8.85 on C10-5T, C100-10T and C100-20T, respectively. iCaRL forgets less than ours on C10-5T and C100-20T as it achieves 4.95 and 8.31, respectively. However, iCaRL was not able to adapt to new knowledge effectively as its accuracies are much lower than our method EWT on the same experiments. The average accuracy over the 6 experiments of EWT is 72.98 while that of iCaRL is only 65.47. According to the forgetting rates, the best baseline (DER) adapts to new knowledge well, but it was not able to retain the knowledge as effectively as our method. Its forgetting rates are 13.36, 15.92, and 15.48 on C10-5T, C100-10T, and C100-20T, respectively, and are much larger than ours. This results in lower average performance of DER than EWT.

## 4.2 Analysis and Ablation

**Performances of Different Variant methods.** Tab. 2 shows the performance gain by adding each proposed technique. The methods in the first (S1) and second rows (S1 + S2) only produce the WTP probability without TIP probability since the TIP is not computed. Thus, we cannot decompose the CIL probability as EWT. Instead, we make a CIL prediction by taking $\arg\max$ over the concatenated logits from each task classifier $g^i$, which is better than the other considered prediction methods (refer to Appendix C). From Tab. 2, fine-tuning the classifier via the generated feature vectors in step 2 already improves the performance from step 1 as shown in the second row (S1 + S2). On C10-5T, C100-10T, and C100-20T, S2 improves more than 3% from S1. When the proposed task-id prediction (TIP) is introduced, the performance also improves as represented in the third row (S1 + TIP). In fact, this is slightly better than S1 + S2 without TIP which implies the effectiveness of the proposed problem decomposition for CIL. Combining all the proposed techniques together delivers the best performance as represented by the last row, which is the full EWT.

**Performance by the Number of Eigenpairs.** Step 1 and 2 are based on generating pseudo feature vectors from Gaussian distributions. Due to the memory consumption, we approximate the covariance by incremental PCA and save only $|\mathcal{M}|$ eigenvectors with the corresponding eigenvalues. This is equivalent to saving $|\mathcal{M}|/C$ eigenpairs per class for a dataset of $C$ classes when learning the last task. Tab. 3 shows the model performance on C100-10T with different $|\mathcal{M}|$ sizes. With a single eigenvector per class (i.e., $|\mathcal{M}| = 100$), the model already achieves 73.41% accuracy. The performance increases with the size of $\mathcal{M}$ until $|\mathcal{M}| = 2,000$. The lower performance on 2,000 is because the less informative eigenpairs now generate noisy feature vectors.

## 5 Conclusion

This paper proposed an effective method to solve class-incremental learning (CIL) from the first principle. Based on the definition of CIL, it first decomposes the CIL prediction probability into two probabilities, within-task prediction (WTP) probability and task-id prediction (TIP) probability. Novel methods are designed to estimate these probabilities, which are based on an incremental PCA-based generative approach to fine-tune the multi-head task classifiers using a single head approach and Mahalanobis distance, respectively. Experimental results show that the proposed EWT outperforms existing strong baselines by a large margin.

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

## A    PSEUDO-CODE

We provide the pseudo-code for training and testing. Our comments start with symbol "//".

---

**Algorithm 1** Training (Step 1)

---

1: **for** training data $D^i$ of each task **do**
2:     **for** each batch $(X_j, Y_j) \subset D^i$, until converge **do**
3:         Obtain features $Z_j = f^i(X_j)$ and outputs $g^i(Z_j)$
            // Compute the mean and eigenpairs for features of each class in the task
4:         **for** each class $c$ feature $Z_c \subset Z_j$ **do**
5:             Compute $\mu_c$ using Eq. 3 and the eigenpairs $(U_c, \Lambda_c)$ using Eq. 5
6:         **end for**
            // Train the classifier with generated features and remember the distributions
7:         Generate pseudo features $\bar{Z}$ from the distributions of the current task and obtain $g^i(\bar{Z})$
8:         Minimize Eq. 7 and update the parameters
9:     **end for**
10: **end for**

---

---

**Algorithm 2** Training (Step 2)

// After training task $t$, fine-tune the classifier
1: Construct $g = [g^1, ..., g^t]$ by concatenating the parameters of each task classifier $g^i$
   // Fine-tuning starts
2: **for** until converge **do**
3:    Generate pseudo feature vectors $\boldsymbol{Z}$ from the distributions of all the learned classes
4:    Minimize Eq. 8 and update the parameters
5: **end for**

---

**Algorithm 3** Testing

**Require:** Test instance $\boldsymbol{x}$, task networks $[f^1, ..., f^t]$, and classifiers $[g^1, ..., g^t]$ after learning task $t$
   // Obtain CIL probabilities of the classes corresponding to each task
1: **for** for each task $i \le t$ **do**
2:    Obtain WTP using Eq. 9 and TIP using Eq. 11
3:    Obtain the CIL probability $p(\boldsymbol{Y}^i|\boldsymbol{x}) = p(\boldsymbol{Y}^i|\boldsymbol{x}, i)p(i|\boldsymbol{x})$, where $\boldsymbol{Y}^i$ is the set of class labels of task $i$
4: **end for**
   // Concatenate the probabilities for the full CIL probability and make a prediction
5: $\hat{y} = \arg\max \bigoplus_i p(\boldsymbol{Y}^i|\boldsymbol{x})$, where $\bigoplus$ is concatenation

---

Table 4: The size of the model (in entries) required for each method without the memory buffer.

| Method | C10-5T | C100-10T | C100-20T | T-5T | T-10T | I380-10T |
|--------|--------|----------|----------|------|-------|----------|
| HAT    | 24.1M  | 24.4M    | 24.7M    | 24.3M | 24.4M | 24.5M    |
| OWM    | 26.6M  | 28.1M    | 28.1M    | 28.2M | 28.2M | 28.3M    |
| SLDA   | 21.6M  | 21.6M    | 21.6M    | 21.7M | 21.7M | 21.7M    |
| PASS   | 22.9M  | 24.2M    | 24.2M    | 24.3M | 24.4M | 24.6M    |
| L2P    | 21.7M  | 21.7M    | 21.7M    | 21.8M | 21.8M | 21.8M    |
| iCaRL  | 22.9M  | 24.1M    | 24.1M    | 24.1M | 24.1M | 24.2M    |
| A-GEM  | 26.5M  | 31.4M    | 31.4M    | 31.5M | 31.5M | 31.6M    |
| EEIL   | 22.9M  | 24.1M    | 24.1M    | 24.1M | 24.1M | 24.2M    |
| DER++  | 22.9M  | 24.1M    | 24.1M    | 24.1M | 24.1M | 24.2M    |
| HAL    | 22.9M  | 24.1M    | 24.1M    | 24.1M | 24.1M | 24.2M    |
| DER    | 27.7M  | 45.4M    | 69.1M    | 33.6M | 45.5M | 45.5M    |
| EWT    | 24.1M  | 24.4M    | 24.7M    | 24.3M | 24.4M | 24.5M    |

## B  REQUIRED MEMORY

We report the network sizes of the systems after learning the last task. We use an 'entry' to denote a parameter or a value required to learn and to inference for a task.

All the systems except SLDA and L2P use the feature extractor DeiT-S/16 (Touvron et al., 2021) and adapter modules. The transformer consumes 21.6 millions (M) entries and the adapters take 1.2M and 2.4M entries for CIFAR10 and the other datasets. SLDA fine-tunes only the classifier on top of the fixed pre-trained feature extractor as it does not have a protection mechanism. L2P uses a prompt pool with 23k entries. Since each method requires method-specific elements (e.g., task embedding for HAT), the number of entries required for each method is different. The number of entries for each model is reported in Tab. 4.

Our method saves the mean and eigenpairs to approximate the distribution of features for each class to draw pseudo feature vectors while the replay-based methods save the raw inputs to replay jointly with the current task data. As the number of eigenpairs and the number of saved inputs affect the performance, we use a budget $\mathcal{M}$ of size $|\mathcal{M}|$ each method can save. Since feature dimension is 384, the total entries required for saving the mean and eigenpairs for our method are 42.2k, 422.4k, 844.8k and 3.1M for CIFAR10, CIFAR100, Tiny-ImageNet, and ImageNet380, respectively. The

Table 5: The methods 1), 2), and 3) indicate entropy-based WTP prediction, softmax-based prediction, and logit-based prediction, respectively, as described Appendix C. The last column Average means the average value over the 6 experiments.

| Method | C10-5T | C100-10T | C100-20T | T-5T | T-10T | I380-10T | Average |
|---|---|---|---|---|---|---|---|
| HAT | | | | | | | |
| 1) | 79.27±5.15 | 66.14±0.33 | 61.32±0.64 | 56.89±1.63 | 55.83±1.53 | 61.26±3.25 | 63.45 |
| 2) | 79.27±5.15 | 68.96±0.29 | 61.83±0.66 | 65.84±0.62 | 62.12±0.46 | 71.16±0.87 | 68.20 |
| 3) | 79.36±5.16 | 68.99±0.21 | 61.83±0.62 | 65.85±0.60 | 62.05±0.55 | 71.20±0.99 | 68.21 |
| S1 | | | | | | | |
| 1) | 73.71±4.06 | 65.05±0.71 | 59.74±0.56 | 55.34±1.66 | 54.03±1.36 | 61.60±2.80 | 61.58 |
| 2) | 73.71±4.06 | 67.89±0.45 | 60.00±0.73 | 64.29±0.57 | 60.55±0.21 | 70.68±0.84 | 66.19 |
| 3) | 74.00±3.93 | 67.76±0.49 | 59.91±0.66 | 64.35±0.61 | 60.48±0.24 | 70.66±0.91 | 66.19 |
| S1 + S2 | | | | | | | |
| 1) | 76.55±4.10 | 67.25±0.43 | 63.90±0.71 | 54.75±1.57 | 53.66±0.82 | 59.40±1.67 | 62.58 |
| 2) | 76.55±4.10 | 71.21±0.47 | 65.05±0.37 | 65.29±0.32 | 61.75±0.17 | 72.32±0.25 | 68.70 |
| 3) | 84.05±2.24 | 71.22±0.44 | 66.11±0.72 | 65.25±0.39 | 61.95±0.47 | 71.95±0.79 | 70.09 |

sizes of raw inputs are 3*32*32, 3*64*64, and 3*224*224 (after resize) for CIFAR, Tiny-ImageNet, and ImageNet380. Thus, the total entries required for memory budget are 614.4k, 6.1M, 24.6M, and 1,144.0M for CIFAR10, CIFAR100, Tiny-ImageNet, and ImageNet380.

Finally, our system saves the covariance matrices for computing TIP in Sec. 3.2.2. The covariances are saved for each task. Since each covariance is in size 384x384, the total entries for this step are 737.3k, 1.5M, 2.9M, 737.3k, 1.5M, and 1.5M for C10-5T, C100-10T, C100-20T, T-5T, T-10T, and I380-10T, respectively. The numbers are relatively small considering that some of the replay-based methods (e.g., iCaRL, HAL) require a teacher model the same size as the training model for knowledge distillation.

## C    DIFFERENT PREDICTION METHODS

As HAT is designed for task incremental learning and does not provide a task-id prediction mechanism as the other parameter isolation methods such as HyperNet (von Oswald et al., 2020), we have tried different CIL prediction methods. The reported values in Tab. 1 are the results of the best one among the considered methods.

We considered three methods:

1) $\arg\max p(\boldsymbol{Y}^i|\boldsymbol{x}, i)$, where the task-id $i$ is chosen based on the entropy values from each task network as HyperNet.

2) $\arg\max[p(\boldsymbol{Y}^1|\boldsymbol{x}, 1); \cdots ; p(\boldsymbol{Y}^t|\boldsymbol{x}, t)]$, where the within-task prediction (WTP) probability is obtained by taking softmax over logits $g^i(f^i(\boldsymbol{x}))$ of task $i$. This is equivalent to using an equal probability for task-id prediction (TIP) probability.

3) $\arg\max[g^1(f^1(\boldsymbol{x})); \cdots ; g^t(f^t(\boldsymbol{x}))]$.

Tab. 5 shows the results of each prediction method. Based on the result, the entropy-based prediction performs the worst. The reason is that the entropy value from each task network is not as informative as other values since it is not trained with entropy. The softmax-based and logit-based predictions are not different on average over the 6 experiments. Since logit-based performance is the best, we choose it as the CIL prediction method for HAT.

S1 and S1+S2 in Tab. 2 in the main paper also do not have CIL prediction mechanism. We try the three prediction methods as HAT. The results are in Tab. 5. We can observe similar behaviors in S1 and S1+S2 as HAT.

Table 6: Incremental classification accuracy. The last column shows the average of accuracies of each method over all the experiments. We highlight the best results in each column in bold.

| Method | C10-5T | C100-10T | C100-20T | T-5T | T-10T | I380-10T | Average |
|--------|--------|----------|----------|------|-------|----------|---------|
| HAT | 87.64±2.63 | 79.26±1.13 | 73.91±0.68 | 74.26±0.66 | 72.61±0.69 | 79.40±1.47 | 77.84 |
| OWM | 56.00±3.46 | 40.10±1.86 | 32.58±1.58 | 45.18±0.33 | 35.75±2.21 | 5.95±4.76 | 35.93 |
| SLDA | **93.54**±0.66 | 77.72±0.58 | 78.51±0.58 | 66.03±1.35 | 67.39±1.81 | 69.66±0.02 | 75.48 |
| PASS | 89.03±7.13 | 77.01±2.44 | 76.42±1.23 | 67.12±6.26 | 67.33±3.63 | 74.76±2.33 | 75.28 |
| L2P | 84.60±2.28 | 72.88±1.18 | 66.52±1.61 | 67.81±1.25 | 64.59±1.59 | 68.09±1.73 | 70.75 |
| iCaRL | 89.74±6.63 | 76.50±3.56 | 77.06±2.36 | 61.36±6.21 | 63.56±3.08 | 73.71±2.13 | 73.65 |
| A-GEM | 68.19±3.24 | 43.83±0.69 | 35.97±1.15 | 49.26±0.64 | 39.58±3.32 | 50.16±6.63 | 47.83 |
| EEIL | 90.50±0.72 | 81.10±0.37 | 79.54±0.69 | 66.63±0.40 | 66.54±0.61 | 75.08±1.07 | 76.57 |
| DER++ | 89.01±6.29 | 80.64±2.74 | 81.72±1.76 | 66.55±3.73 | 67.14±1.40 | 77.41±0.37 | 77.08 |
| HAL | 87.00±7.27 | 77.42±2.73 | 77.85±1.71 | 65.31±3.68 | 64.48±1.45 | 75.87±0.40 | 74.65 |
| DER | 92.83±1.10 | **82.89**±0.45 | **82.79**±0.76 | 70.32±0.57 | 70.21±0.86 | 78.30±0.67 | 79.56 |
| EWT | 93.20±1.84 | 82.57±0.69 | 80.52±0.85 | **74.27**±0.70 | **73.87**±1.00 | **81.24**±1.65 | **80.94** |

## D   ADDITIONAL DERIVATION DETAILS

We have claimed that the orthonormal matrix $\tilde{U}$ and a diagonal matrix $\tilde{\Lambda}^2$ obtained from Eq. 5 in the main text are eigenvectors and eigenvalues of unnormalized sample covariance $(n + m - 1)\Sigma$. Denote the previous sample mean by $\mu$ and the eigenpairs of previous covariance $\Sigma_{\text{old}}$ by $(U, \Lambda^2)$. Following Ross et al. (2008), we provide more details about the claim. Since $\hat{K} = [X_{\text{new}} - \mu_{\text{new}} \quad \sqrt{nm/(n+m)}(\mu_{\text{new}} - \mu)]$ and $U\Lambda U^T = \Sigma_{\text{old}}$,

$$\tilde{U}\tilde{\Lambda}^2\tilde{U}^T = \tilde{U}\tilde{\Lambda}\tilde{V}^T[\tilde{U}\tilde{\Lambda}\tilde{V}^T]^T \tag{12}$$

$$= [\sqrt{n-1}U\Lambda \quad \hat{K}][\sqrt{n-1}U\Lambda \quad \hat{K}]^T \tag{13}$$

$$= (n-1)U\Lambda^2 U^T + \hat{K}\hat{K}^T \tag{14}$$

$$= (n-1)\Sigma_{\text{old}} + (m-1)\Sigma_{\text{new}} \tag{15}$$

$$\quad + \frac{nm}{n+m}(\mu_{\text{new}} - \mu)(\mu_{\text{new}} - \mu)^T$$

$$= (n+m-1)\Sigma \tag{16}$$

where the last derivation from Eq. 15 to Eq. 16 is by Lemma 1 of Ross et al. (2008).

## E   INCREMENTAL CLASSIFICATION ACCURACY

In the main paper, we reported the average classification accuracy (ACA) after learning the last task. In this section, we also report the incremental classification accuracy (ICA) over the learning process. ICA after task $t$ is defined as $\bar{A}_t = \sum_{i=1}^{t} A_i$, where $A_i$ is ACA after learning task $i$. Tab. 6 shows ICA of our method EWT and the baselines. For the more challenging datasets (e.g., Tiny-ImageNet and ImageNet), our system outperforms the baselines. SLDA is slightly better than our method on C10-5T and DER is slightly better than EWT on C100-10T and 20T. However, their performances are not consistent over different experiments. The average performance of our method over the 6 experiments is 80.94 while the best performing baseline DER is 79.56. Fig. 3 shows the ACA at each task.

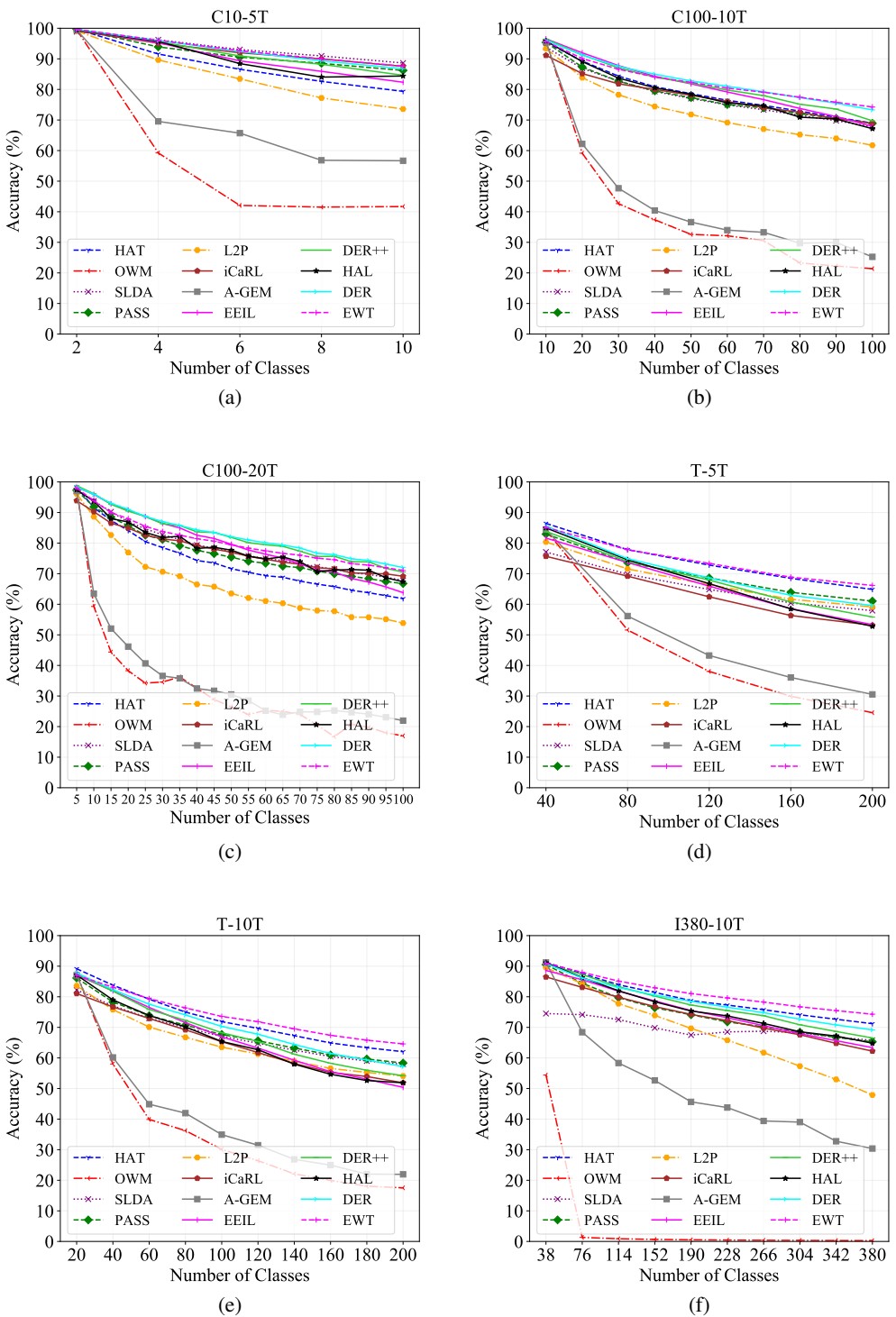

Figure 3: Average classification accuracy after each task. The x-axis indicates the number of learned classes after each task. The systems with dashed lines are exemplar-free methods.

