# OpenReview forum: "Solving Continual Learning via Problem Decomposition"
_ICLR.cc/2023/Conference — Submitted to ICLR 2023_

### Official Review · Reviewer_D1ii · 2022-10-23

**Confidence:** 4
**Correctness:** 3
**Technical Novelty And Significance:** 2
**Empirical Novelty And Significance:** 2
**Recommendation:** 5

**Clarity, Quality, Novelty And Reproducibility:**

I think the decomposition of CIL proposed in this paper is good, and the authors designed some significant strategies to solve it. But these  strategies are based on existing methods and may rely heavily on pre-trained models. The novelty is not enough.

**Strength And Weaknesses:**

Strength:
1. This paper proposed a new perspective to decompose the CIL probability into WTH and TIP probability.
2. For the above two sub-problems, the paper proposes a simple yet effective solution based on HAT, PCA, and features replay.
3. The proposed EWT outperforms the existing methods without exemplar replay.

Weaknesses:
1. The pre-trained model used in the experiments still too strong for CIFAR and Tiny-/Sub-ImageNet.
2. The proposed methods could be highly dependent on the good feature representation obtained by pre-trained model. If not,  some experiments under normal settings are needed to prove it.
3. There is no quantitatively comparison of additional storage between EWT and other replay-based methods.

**Summary Of The Paper:**

This paper decomposes the CIL problem into two prediction steps, which are within-task prediction and task-id prediction. Several strategies are designed to tackle these sub-problems, e.g. HAT, PCA, single head classifier fine-tuning. Experimental results show that the proposed EWT works better than existing baselines on pre-trained transformer network.

**Summary Of The Review:**

Based on the above comments, I tend to reject this paper. The main reason is that the experimental settings are insufficient to prove the effectiveness of the method.

---

> ### Author Response · Authors · 2022-11-15
> **Author response to Reviewer D1ii**
>
> 1. The pre-trained model used in the experiments still too strong for CIFAR and Tiny-/Sub-ImageNet.
>
> 	**Response**: Please see the response to Comment 2.
>
> 2. The proposed methods could be highly dependent on the good feature representation obtained by pre-trained model. If not, some experiments under normal settings are needed to prove it.
>
> 	**Response**: We use a pre-trained model as pre-trained models have been widely used in CL [e.g., 1, 2, 3, 4, 5, 6] recently as strong pre-trained models are proposed. In our experiments, we used 611 classes of ImageNet after removing 389 classes similar or identical to the classes in the CIFAR and Tiny-ImageNet datasets to make sure there is no information leak from the pre-trained model to CL.
>
> 	To validate the effectiveness of our proposed method with a weaker feature extractor, we have conducted additional experiments. The new feature extractor is pre-trained using only 200 randomly selected classes from the 611 classes of ImageNet after removing 389 classes similar or identical to the classes in the CIFAR and Tiny-ImageNet datasets. We implemented our method EWT and the 5 best performing baselines (HAT, SLDA, PASS, DER++, and DER). The results are as follows.
>
> 	|       | C10-5T | C100-10T | C100-20T | T-5T  | T-10T | I380-10T | Avg.  |
> 	| ------| -------| -------- | -------- | ----- | ----- | -------- | ----- |
> 	| HAT   | 73.52  |   57.12  |   49.44  | 52.48 | 47.64 |   58.00  | 56.37 |
> 	| SLDA  | **79.93**  |   55.59  |   55.66  | 43.64 | 43.60 |   48.54  | 54.49 |
> 	| PASS  | 76.92  |   57.67  |   54.64  | 48.51 | 45.01 |   49.18  | 55.32 |
> 	| DER++ | 69.11  |   56.02  |   56.22  | 40.67 | 38.51 |   24.98  | 47.59 |
> 	| DER   | 73.79  |   59.75  |   **58.10**  | 46.81 | 43.80 |   60.95  | 57.20 |
> 	| EWT   | 77.35  |   **62.57**  |   57.33  | **53.70** | **50.93** |   **62.41**  | **60.72** |
>
> 	SLDA performs well in C10-5T. However, it performs poorly on more challenging datasets as it freezes the feature extractor and cannot adapt to new knowledge. The best-performing baseline DER achieves an average of 57.20 over the 6 experiments while we achieve 60.72. Thus, our proposed method EWT still outperforms the baselines even with a weaker feature extractor.
>
> **** Continued on the next response ****

---

> > ### Author Response · Authors · 2022-11-15
> > **Author response to Reviewer D1ii**
> >
> > 3. There is no quantitatively comparison of additional storage between EWT and other replay-based methods.
> >
> > 	**Response**: This is a misunderstanding. We discussed the required storage/memory in Footnote 5 of the main paper and referred to Appendix B for the memory storage required for all the systems including ours and the replay-based methods. We reproduce Appendix B here for the five best performing baselines. Please refer to Appendix B for the full results. We use 'entry' to indicate a parameter or a value required at training or inference.
> >
> > 	The systems require the following number of entries for the network without including the replay/memory buffer. The unit for each number is in million (M).
> >
> > 	|       | C10-5T | C100-10T | C100-20T | T-5T  | T-10T | I380-10T |
> > 	| ------| -------| -------- | -------- | ----- | ----- | -------- |
> > 	| HAT   | 24.1   |   24.4   |   24.7   | 24.3  | 24.4  |   24.5   |
> > 	| SLDA  | 21.6   |   21.6   |   21.6   | 21.7  | 21.7  |   21.7   |
> > 	| PASS  | 22.9   |   24.2   |   24.2   | 24.3  | 24.4  |   24.6   |
> > 	| DER++ | 22.9   |   24.1   |   24.1   | 24.1  | 24.1  |   24.2   |
> > 	| DER   | 27.7   |   45.4   |   69.1   | 33.6  | 45.5  |   45.5   |
> > 	| EWT   | 24.1   |   24.4   |   24.7   | 24.3  | 24.4  |   24.5   |
> >
> > 	The network sizes of our method are better or competitive to the best-performing replay-based baselines (DER++ and DER). The exemplar-free methods HAT, SLDA, and PASS require less or similar memory than the proposed method EWT, but their performances are much lower. Each system achieves 68.21, 67.65, and 67.75% accuracy, respectively over the 6 experiments, while EWT achieves 72.98.
> >
> > 	Regarding the replay buffer/memory and our saved information, we analyze as follows. Our method saves the mean and eigenpairs to approximate the distribution of features for each class while the replay-based methods save the raw inputs for replay. As the number of eigenpairs in our method and the number of saved inputs in the replay buffer affect the performance, we fix the budget size of how many eigenpairs or raw data the system can save. With the budget size reported in Training Details in Section 4 of the main paper, the total entries required for saving the mean and eigenpairs for our method are 42.2 thousand (K), 422.4K, 844.8K, 3.1M for CIFAR10, CIFAR100, Tiny-ImageNet, and ImageNet380, respectively. The total entries required for the replay buffer/memory budgets in the replay methods are 614.4K, 6.1M, 24.6M, and 1.1 billion (B), respectively.
> >
> > 	Finally, our system saves covariance matrices for computing task-id prediction. The covariances are saved for each task. The total entries for this step are 737.3K, 1.5M, 2.9M, 737.3K, 1.5M, and 1.5M for C10-5T, C100-10T, C100-20T, T-5T, T-10T, and I380-10T, respectively.
> >
> > 	In summary, our system requires much less memory than the replay-based baselines in total and similar memory to the exemplar-free baselines despite that it outperforms the baselines. Please refer to Appendix B for more detailed discussion.
> >
> > 4. I think the decomposition of CIL proposed in this paper is good, and the authors designed some significant strategies to solve it. But these strategies are based on existing methods and may rely heavily on pre-trained models. The novelty is not enough.
> >
> > 	**Response**: The mechanism for preventing catastrophic forgetting is based on the existing method HAT. However, the core idea of solving the CIL problem by the decomposition is principled and novel. Moreover, the proposed method outperforms the strong baselines with both strong and weak feature extractors.
> >
> > 5. Based on the above comments, I tend to reject this paper. The main reason is that the experimental settings are insufficient to prove the effectiveness of the method.
> >
> > 	**Response**: Please see our response to Comment 2.
> >
> > [1] Fei Zhu, Xu-Yao Zhang, Chuang Wang, Fei Yin, and Cheng-Lin Liu. Prototype augmentation and self-supervision for incremental learning. CVPR, 2021.
> >
> > [2] Shipeng Yan, Jiangwei Xie, and Xuming He. DER: Dynamically expandable representation for class incremental learning. CVPR, 2021.
> >
> > [3] Zixuan Ke, Bing Liu, Nianzu Ma, Hu Xu, and Lei Shu. Achieving forgetting prevention and knowledge transfer in continual learning. NeurIPS, 2021.
> >
> > [4] Oleksiy Ostapenko, Timothee Lesort, Pau Rodŕıguez, Md Rifat Arefin, Arthur Douillard, Irina Rish, and Laurent Charlin. Continual learning with foundation models: An empirical study of latent replay. CoLLAs, 2022.
> >
> > [5] Zifeng Wang, Zizhao Zhang, Chen-Yu Lee, Han Zhang, Ruoxi Sun, Xiaoqi Ren, Guolong Su, Vincent Perot, Jennifer Dy, and Tomas Pfister. Learning to prompt for continual learning. CVPR, 2022.
> >
> > [6] Tz-Ying Wu, Gurumurthy Swaminathan, Zhizhong Li, Avinash Ravichandran, Nuno Vasconcelos, Rahul Bhotika, and Stefano Soatto. Class-incremental learning with strong pre-trained models. CVPR, 2022.

---

> ### Author Response · Authors · 2022-11-22
> **A gentle reminder**
>
> Dear Reviewer,
>
> We are wondering whether we have answered your questions satisfactorily. Our paper proposes a principled method to solve the class-incremental learning problem, which is very different from existing approaches.
>
> Thank you

---

> ### Author Response · Authors · 2022-12-08
> **A reminder for the end of the discussion period**
>
> Dear Reviewer,
>
> As the end of the discussion period is coming (Dec. 12), we are wondering if your concerns have been addressed by our responses. We’d like to emphasize that we provide a principled approach to CIL, and the proposed techniques are effective in achieving a good CIL performance. Please let us know if you have any questions. We are happy to answer them.
>
> Thank you

---

### Official Review · Reviewer_cHhV · 2022-10-25

**Confidence:** 3
**Correctness:** 3
**Technical Novelty And Significance:** 2
**Empirical Novelty And Significance:** 2
**Recommendation:** 8

**Clarity, Quality, Novelty And Reproducibility:**

The problem setting is clearly defined and the method is well explained.
It is hard to judge the novelty of the method because, while related work is referenced, it is not described, thus, the difference from similar past work is not clear.

**Strength And Weaknesses:**

I think that the method is comprised of insights which could be of interest to the community. Moreover, the experiment compare the method many baselines and conduct ablation studies.
One potential weakness is that the approach relies on a pre-trained backbone. Another issue is that the “Related Work” section does not describe and compare to the most closely related work, but rather just cites it.


**Summary Of The Paper:**

This paper considers class incremental continual learning, in which disjoint subsets of classes are introduced one at a time. A multi-head architecture is used which shares a pre-trained feature extractor (backbone). Forgetting is addressed by using HAT (Serra et al., 2018) which applies a separate binary mask to the backbone for each task (subset of classes).
The authors decompose the prediction into task-id prediction (TIP) probability and within-task prediction (WTP) probability. Both are estimated by making use of class-specific approximations of the features distribution.
The resulting method is shown to outperform competitive baselines and the design decisions are motivated with ablation experiments.


**Summary Of The Review:**

The paper present  interesting ideas and satisfactory experiments. Conversely, I am uncertain about the novelty of the method.

---

> ### Author Response · Authors · 2022-11-15
> **Author response to Reviewer cHhV**
>
> 1. I think that the method is comprised of insights which could be of interest to the community. Moreover, the experiment compare the method many baselines and conduct ablation studies. One potential weakness is that **(1.1) the approach relies on a pre-trained backbone**. Another issue is that **(1.2) the “Related Work” section does not describe and compare to the most closely related work, but rather just cites it**.
>
> 	**Response to Comment 1.1**: We use a pre-trained model as pre-trained models have been widely used in CL [e.g., 1, 2, 3, 4, 5, 6] recently as strong pre-trained models are proposed.
>
> 	To validate the effectiveness of our proposed method with a weaker feature extractor, we have conducted additional experiments. The new feature extractor is pre-trained using only 200 randomly selected classes from the 611 classes of ImageNet after removing 389 classes similar or identical to the classes in the CIFAR and Tiny-ImageNet datasets. We implemented our method EWT and the 5 best performing baselines. The results are as follows.
>
> 	|       | C10-5T | C100-10T | C100-20T | T-5T  | T-10T | I380-10T | Avg.  |
> 	| ------| -------| -------- | -------- | ----- | ----- | -------- | ----- |
> 	| HAT   | 73.52  |   57.12  |   49.44  | 52.48 | 47.64 |   58.00  | 56.37 |
> 	| SLDA  | **79.93**  |   55.59  |   55.66  | 43.64 | 43.60 |   48.54  | 54.49 |
> 	| PASS  | 76.92  |   57.67  |   54.64  | 48.51 | 45.01 |   49.18  | 55.32 |
> 	| DER++ | 69.11  |   56.02  |   56.22  | 40.67 | 38.51 |   24.98  | 47.59 |
> 	| DER   | 73.79  |   59.75  |   **58.10**  | 46.81 | 43.80 |   60.95  | 57.20 |
> 	| EWT   | 77.35  |   **62.57**  |   57.33  | **53.70** | **50.93** |   **62.41**  | **60.72** |
>
> 	The average accuracy of our method over the 6 experiments is 60.72 while the best performing baseline DER is 57.20. All the systems experience a drop in performance as a weaker feature extractor is used. However, our method still outperforms the baselines.
>
> 	**Response to Comment 1.2**: Thank you for carefully reading it. We tried to explain the similarities and differences, but overlooked some details. We have revised and emphasized the differences more clearly. Please check the related work section in the revised paper.
>
> 2. The problem setting is clearly defined and the method is well explained. It is hard to judge the novelty of the method because, while related work is referenced, it is not described, thus, the difference from similar past work is not clear.
>
> 	**Response**: Please see our response to Comment 1.2. Moreover, the core idea of solving the CIL problem by the decomposition is principled and novel. Several methods are proposed to estimate the probabilities and are shown to be effective in extensive experiments.
>
> [1] Fei Zhu, Xu-Yao Zhang, Chuang Wang, Fei Yin, and Cheng-Lin Liu. Prototype augmentation and self-supervision for incremental learning. CVPR, 2021.
>
> [2] Shipeng Yan, Jiangwei Xie, and Xuming He. DER: Dynamically expandable representation for class incremental learning. CVPR, 2021.
>
> [3] Zixuan Ke, Bing Liu, Nianzu Ma, Hu Xu, and Lei Shu. Achieving forgetting prevention and knowledge transfer in continual learning. NeurIPS, 2021.
>
> [4] Oleksiy Ostapenko, Timothee Lesort, Pau Rodŕıguez, Md Rifat Arefin, Arthur Douillard, Irina Rish, and Laurent Charlin. Continual learning with foundation models: An empirical study of latent replay. CoLLAs, 2022.
>
> [5] Zifeng Wang, Zizhao Zhang, Chen-Yu Lee, Han Zhang, Ruoxi Sun, Xiaoqi Ren, Guolong Su, Vincent Perot, Jennifer Dy, and Tomas Pfister. Learning to prompt for continual learning. CVPR, 2022.
>
> [6] Tz-Ying Wu, Gurumurthy Swaminathan, Zhizhong Li, Avinash Ravichandran, Nuno Vasconcelos, Rahul Bhotika, and Stefano Soatto. Class-incremental learning with strong pre-trained models. CVPR, 2022.

---

### Official Review · Reviewer_vcxy · 2022-10-25

**Confidence:** 2
**Correctness:** 1
**Technical Novelty And Significance:** 3
**Empirical Novelty And Significance:** 3
**Recommendation:** 3

**Clarity, Quality, Novelty And Reproducibility:**

*Clarity
- The subscripts for w_k,j,l are not defined.
- The motivation for the regularization in Eq.7 is unclear.
- Why do they use the OOD samples in the computation of WTP?
- It is better to show the results after each task as well as the average performance over them. The authors only provide average classification accuracy after the final task.

*Quality
- Please see the above comments.

*Novelty
- The proposed method seems to be novel.

*Reproducibility
- Code is available.

**Strength And Weaknesses:**

*Strength
- Experimental results of the proposed method showed promising performance.

*Weaknesses
- The important assumption of why the class labels of tasks are disjoint is not well justified.
- The problem setting is unclear. It is described in about four lines in the Definition paragraph but without a specific flow. It is difficult to understand what they actually do. For example, I cannot see when task IDs are estimated, when tasks are switched, tasks come in altogether; they come once, they don't come again, or they come repeatedly. Will all data be discarded each time, or will it be discarded for each task?
- The motivation of Eq.2 is unclear. Why do the authors introduce pseudo feature vectors Z? The description of the method is also unclear. How can we generate y for Z? Where does B come from?

**Summary Of The Paper:**

This paper addresses continual learning where task IDs are not available.
The authors propose a learning method preventing catastrophic forgetting and providing task ID estimation capability.
The proposed method utilizes batches of features for each task generated from the generators obtained with incremental PCA. At this time, the OOD generator learned from past tasks is also used. The method estimates the ID for the current sample as that of the closest task in terms of the Mahalanobis distance from the mean and variance of the distribution of features of each task.
Experimental results on multiple public datasets demonstrate that the proposed method performs better than existing methods.

**Summary Of The Review:**

The clarity is low. The motivation and details of the proposed method are unclear.

---

> ### Author Response · Authors · 2022-11-15
> **Author response to Reviewer vcxy**
>
> 1. The important assumption of why the class labels of tasks are disjoint is not well justified.
>
> 	**Response**: There are several settings of continual learning. We follow the standard definition of class-incremental learning (CIL) [1, 2, 3, 4], where tasks do not have overlapping classes or tasks have disjoint classes. As we noted in Footnote 2, if a learned class in a task appears again later, we can define tasks as follows: Suppose dataset 1 has classes {dog, cat, tiger} and dataset 2 has classes {dog, computer, car}. We can define task 1 as {dog, cat, tiger} and task 2 as {computer, car}. The shared class dog in dataset 2 can be regarded as additional training data of the dog that have appeared after task 1. Our work is based on offline CIL, which assumes all the training data of a task arrives together. In the online CIL, the above situation can happen, i.e., some data of a class in a task may appear again later together with the data of another task, which is called the blurry task boundary setting [5].
>
> 2. The problem setting is unclear. It is described in about four lines in the Definition paragraph but without a specific flow. It is difficult to understand what they actually do. For example, I cannot see when task IDs are estimated, when tasks are switched, tasks come in altogether; they come once, they don't come again, or they come repeatedly. Will all data be discarded each time, or will it be discarded for each task?
>
> 	**Response**: There may be a major misunderstanding here. We follow the standard definition of CIL [1, 2, 3, 4]. Also, as we stated in a few places (e.g., the second paragraph of page 2, the first paragraph of page 4, and the title of Section 3.2), our work does not estimate task id. Instead, we estimate the two probabilities of Eq. 1 (Section 3.2.1 and Section 3.2.2).
>
> 	Regarding task switching, we also follow the standard offline CIL setting [1, 2, 3, 4], where the training data of each task comes together once and will not come again. In the online CIL, the classes of each task may come again and again [5], which is called the revisiting case or blurry task boundary case. This is discussed and compared with the learning scenario in this work in the last paragraph of Related Work. This revisiting case or blurry task boundary is under-studied in offline CIL perhaps due to the difficulty in experiment set-up. As discussed in Footnote 2, we leave this scenario for our future work. Our work is not about online CL.
>
> 	About whether we discard the data of each task after training, yes. As discussed in (e.g., abstract, the second last paragraph in Introduction in page 2, and first and second paragraphs in Related Work), we do not save any replay data.
>
> 3. The motivation of Eq.2 is unclear. Why do the authors introduce pseudo feature vectors Z? The description of the method is also unclear. How can we generate y for Z? Where does B come from?
>
> 	**Response**: As we stated in the first paragraph of Section 3.1, “we use Z in the step 1 training because we want to produce better distributions, which will be used in step 2.” The label y for Z is known since the estimated distributions are labeled. For instance, a generated feature $z$ from distribution $N(\mu_c, \Sigma_c)$ will be labeled with class c. The notation B in Eq. 2 is a batch of the task data. The explanation was neglected. We have fixed it and added an explanation. Please check Eq. 2 in the revised paper.
>
> 4. The subscripts for w_k,j,l are not defined.
>
> 	**Response**: Thank you for pointing it out. We denote a layer by $l$. $k$ and $j$ indicate $k$th row and $j$th column of the weight matrix. We have revised it. Please check Eq. 6 in the revised paper.
>
> 5. The motivation for the regularization in Eq.7 is unclear.
>
> 	**Response**: The regularization in Eq.7 is directly from the hard attention mask (HAT) paper [6] as this part is about the mask. The regularization is employed to encourage parameter sharing between tasks. A thorough study on the role of the regularization is given in [6].
>
> 6. Why do they use the OOD samples in the computation of WTP?
>
> 	**Response**: This may be a misunderstanding. We do not use out-of-distribution (OOD) samples from any external sources. As stated in the first paragraph in Section 3.2.1, in computing WTP, the probability can be inaccurate due to the over-confidence problem of neural networks on anomaly samples (e.g., classes not belonging to the task), which negatively affects the CIL probability. We thus consider the generated features/representations from the distributions of the other tasks as possible OOD data to fine-tune the classifier of a task to alleviate the problem. The effect of this step is shown in the ablation study in Table 2. It improves the performance a great deal. For instance, the CIL accuracy improved from 74.00 to 84.05 on CIFAR10-5T.
>
> **** Continued on the next response ****

---

> > ### Author Response · Authors · 2022-11-15
> > **Author response to Reviewer vcxy**
> >
> > 7. It is better to show the results after each task as well as the average performance over them. The authors only provide average classification accuracy after the final task.
> >
> > 	**Response**: We have revised the paper and reported the accuracies after each task and the average performance over them (i.e. incremental classification accuracy (ICA)) of our proposed method EWT and all the baselines. Please check Figure 3 for the results at each task and Table 6 for ICA in Appendix E in the revised paper. Our method also outperforms the baselines.
> >
> >
> >
> > [1] van de Ven, G. M. and Tolias, A. S. Three scenarios for continual learning. arXiv preprint arXiv:1904.07734, 2019.
> >
> > [2] Marc Masana, Xialei Liu, Bartlomiej Twardowski, Mikel Menta, Andrew D Bagdanov, and Joost van de Weijer. Class-incremental learning: survey and performance evaluation on image classification. arXiv preprint arXiv:2010.15277, 2020
> >
> > [3] Saihui Hou, Xinyu Pan, Chen Change Loy, Zilei Wang, and Dahua Lin. Learning a unified classifier incrementally via rebalancing. CVPR, 2019.
> >
> > [4] Arthur Douillard, Matthieu Cord, Charles Ollion, Thomas Robert, and Eduardo Valle. Podnet: Pooled outputs distillation for small-tasks incremental learning. ECCV, 2020.
> >
> > [5] Jihwan Bang, Heesu Kim, YoungJoon Yoo, Jung-Woo Ha, and Jonghyun Choi. Rainbow memory: Continual learning with a memory of diverse samples. CVPR, 2021.
> >
> > [6] Joan Serra, Didac Suris, Marius Miron, and Alexandros Karatzoglou. Overcoming catastrophic forgetting with hard attention to the task. ICML, 2018.

---

> ### Author Response · Authors · 2022-11-22
> **A gentle reminder**
>
> Dear Reviewer,
>
> We are wondering whether we have answered your questions satisfactorily. Our paper proposes a principled method to solve the class-incremental learning problem, which is very different from existing approaches.
>
> Thank you

---

> ### Author Response · Authors · 2022-12-08
> **A reminder for the end of the discussion period**
>
> Dear Reviewer,
>
> As the end of the discussion period is coming (Dec. 12), we are wondering if your concerns have been addressed by our responses. We’d like to emphasize that we provide a principled approach to CIL, and the proposed techniques are effective in achieving a good CIL performance. Please let us know if you have any questions. We are happy to answer them.
>
> Thank you

---

### Official Review · Reviewer_JguF · 2022-10-25

**Confidence:** 4
**Correctness:** 3
**Technical Novelty And Significance:** 3
**Empirical Novelty And Significance:** 3
**Recommendation:** 6

**Clarity, Quality, Novelty And Reproducibility:**

The proposed approach, which is conceptually simple, is well justified, described and motivated. The paper is well written and experiments are comprehensive. Novelty is limited by similar approaches using distances over summaries in representation space. The authors facilitate reproducibility by providing a detailed account of the model, optimization, experimental settings and source code.

**Strength And Weaknesses:**

The authors propose a relatively simple approach to address CIL tasks via estimating features for each task using iPCA and hard attention masking (i.e., the HAT core) to prevent catastrophic forgetting. The approach simple and intuitive, computationally efficient and leverages the hard attention mechanism in HAT as a means to mitigate catastrophic forgetting.

As a drawback, the authors seem to omit the existing literature in expansion-based approaches for continual learning. Further, expansion-based approaches have demonstrated excellent performance (at the time, so of which, do not consider transformer architectures) in terms of catastrophic forgetting at the expense of moderate parameter growth. For instance, the authors do not mention Mehta et al. 2021, a nonparametric network-expansion approach that also uses a Gaussian distribution and a likelihood similarity (with a Mahalanobis distance calculation at its core) to predict task labels. The latter, though probably will not be able to compete in terms of performance, underscores that the idea of using a summary in representation learning space and a similarity function as a means to identify tasks is not new, thus reducing the novelty of the proposed approach.

The experimental results are extensive and convincing, specially the fact that all baselines use the same architecture and that the transformer was trained to exclude classes similar to CIFAR and Tiny-ImageNet. The details of the experiments are clear and the ablation studies in Table 2 and 3 help understand the contribution of each component of the proposed approach.

As a minor note, Figure 2 is hard to see, so changing the visualization or simply replacing it by a Table will improve the presentation.

**Summary Of The Paper:**

The authors proposed an approach for class incremental learning by leveraging the conditional of the labels given the data (covariates), which they decompose in terms of the labels given the task (consistent with task incremental learning) and the task given the data (task predictor), this without the need of a replay buffer or pseudo-rehearsal. The task prediction is realized incrementally via independent principal component analysis and quantified via Mahalanobis distance.

**Summary Of The Review:**

The authors propose a conceptually simple approach for CIL based on incrementally estimated summaries over previous tasks via iPCA and the hard attention masking of HAT. The resulting approach is computationally efficient, leverages a pre-trained transformer encoder and delivers excellent performance.

---

> ### Author Response · Authors · 2022-11-15
> **Author response to Reviewer JguF**
>
> 1. As a drawback, the authors seem to omit the existing literature in expansion-based approaches for continual learning. Further, expansion-based approaches have demonstrated excellent performance (at the time, so of which, do not consider transformer architectures) in terms of catastrophic forgetting at the expense of moderate parameter growth. For instance, the authors do not mention Mehta et al. 2021, a nonparametric network-expansion approach that also uses a Gaussian distribution and a likelihood similarity (with a Mahalanobis distance calculation at its core) to predict task labels. The latter, though probably will not be able to compete in terms of performance, underscores that the idea of using a summary in representation learning space and a similarity function as a means to identify tasks is not new, thus reducing the novelty of the proposed approach.
>
> 	**Response**: Thank you for suggesting the interesting paper. We have tried to include diverse methods, which cover one expansion-based replay method DER, but overlooked (Mehta et al. 2021) (IBP-WF). We have cited and discussed it in page 3 in the Related Work section. However, the code of IBP-WF is not released to the public, so we are unable to run it.
>
> 	As we discussed in the related work and the experiment sections, there have been multiple attempts to find task-id to solve CIL. Besides IBP-WF, some representative systems are HyperNet, CCG, iTAML, SupSup, and PR (see footnote 3). These models find the task-id explicitly and then use the task-id to select the correct model to make a within-task prediction. Our method is different as we directly compute the CIL probability via the proposed decomposition and estimation of the two probabilities in Eq 1. Also, Table 5 of Appendix C shows that task-id prediction (e.g., entropy-based prediction as in HyperNet or PR) is less accurate. Footnote 3 also shows that the existing task-id prediction methods perform poorly.
>
> 2. The experimental results are extensive and convincing, specially the fact that all baselines use the same architecture and that the transformer was trained to exclude classes similar to CIFAR and Tiny-ImageNet. The details of the experiments are clear and the ablation studies in Table 2 and 3 help understand the contribution of each component of the proposed approach.
> As a minor note, Figure 2 is hard to see, so changing the visualization or simply replacing it by a Table will improve the presentation.
>
> 	**Response**: We have revised it. Please check Figure 2 in the revised paper.
>
> Mehta et al. (2021) Continual learning using a bayesian nonparametric dictionary of weight factors. In International Conference on Artificial Intelligence and Statistics. PMLR, 2021.

---

### Decision · Program_Chairs · 2023-01-20

**Decision:**

Reject

**Justification For Why Not Higher Score:**

Mostly the reject decision is based on the lack of motivation for key technical decisions. To some extent, I share the concerns about the similarity to previous work, but that is not as a major a concern; a combination of simple ideas can be useful if we understand why they should work, and they work well.

**Justification For Why Not Lower Score:**

N/A

**Metareview: Summary, Strengths And Weaknesses:**

This paper presents a framework for class-incremental continual learning in which for each class, the method incrementally maintains a distribution over activations. The class is then predicted by combining the scores of the within-task classifier with a class-conditional Gaussian classifier in activation space, to predict the task.

Strengths:
* Several of the reviewers praised the experimental evaluation, which compares to a large variety of competing methods.
* The combination of incrementally learning the class mean activations and TIP prediction seems to be novel.

Weaknesses:
* A few reviewers mentioned the strong similarity to previous work (e.g., HAT)

* Key design decisions of the approach are not well motivated:

 - For example, the network is separated between a feature extractor f_i, and a network g_i that is specific to each task. It is unclear whether f_i is shared across all the tasks (in which case, why subscript?) or if they are separate to each task (in which case, why separate f from g?). The paper says "Note
that although f_i’s are task specific but they are all learned in the same network and there are a lot of parameter sharing." which I am unable to understand. This issue is important for understanding why we expect HAT to be helpful here.

- I can understand why we want classifiers to be robust in general, but why is this problem more severe in the context of CIL? Why is the class-conditional Gaussian expected to be more robust to OOD data than simply predicting the task id using a separate network (perhaps with shared weights?)

- I share the reviewer's question about the use of synthetic data in task 1. The paper says "we use Z in the step 1 training because we want to produce better distributions", but I think that the question is *why* do we believe that incorporating pseudo feature vectors generated in this way will help learning. Is this simply a way of doing data augmentation? I believe that p(y | z, i) in (2) is undefined. I assume that this is just  p(y |z, i) = g_i(z), but this should be stated.

* As a minor point, although the overall setup of CIL is described in detail in previous work, I do sympathize with the comments of Reviewer vcxy that the description of CIL should be expanded slightly to make the paper more self-contained and help to motivate the method. e.g., make clear that the task boundaries are known, and when the final data arrives, we want to classify based on all the classes seen so far, which can be decomposed into task + class prediction. For example, the paper does not actually say, as far as I can tell, that the method is expected to process all of the training data from each task, and then move on to the next task. However, this did not play a major role in my decision.